# A Lightweight Feature Distillation and Enhancement Network for Super-Resolution Remote Sensing Images

**DOI:** 10.3390/s23083906

**Published:** 2023-04-12

**Authors:** Feng Gao, Liangliang Li, Jiawen Wang, Kaipeng Sun, Ming Lv, Zhenhong Jia, Hongbing Ma

**Affiliations:** 1College of Information Science and Engineering, Xinjiang University, Urumqi 830046, China; 2Key Laboratory of Signal Detection and Processing, Xinjiang University, Urumqi 830046, China; 3Department of Electronic Engineering, Tsinghua University, Beijing 100084, China; 4Shanghai Institute of Satellite Engineering, Shanghai 201109, China

**Keywords:** remote sensing, single image super-resolution, convolutional neural network, feature distillation and enhancement

## Abstract

Super-resolution (SR) images based on deep networks have achieved great accomplishments in recent years, but the large number of parameters that come with them are not conducive to use in equipment with limited capabilities in real life. Therefore, we propose a lightweight feature distillation and enhancement network (FDENet). Specifically, we propose a feature distillation and enhancement block (FDEB), which contains two parts: a feature-distillation part and a feature-enhancement part. Firstly, the feature-distillation part uses the stepwise distillation operation to extract the layered feature, and here we use the proposed stepwise fusion mechanism (SFM) to fuse the retained features after stepwise distillation to promote information flow and use the shallow pixel attention block (SRAB) to extract information. Secondly, we use the feature-enhancement part to enhance the extracted features. The feature-enhancement part is composed of well-designed bilateral bands. The upper sideband is used to enhance the features, and the lower sideband is used to extract the complex background information of remote sensing images. Finally, we fuse the features of the upper and lower sidebands to enhance the expression ability of the features. A large number of experiments show that the proposed FDENet both produces less parameters and performs better than most existing advanced models.

## 1. Introduction

To recover high-resolution (HR) images from low-resolution (LR) images, the single-image super-resolution task (SISR) is a hot topic in computer vision, which is closely related to various computer vision tasks, such as target detection [1,2] and scene marking [3,4].

SISR is an ill-posed problem—that is, there are many solution spaces for restoring HR images from LR images. From the current point of view, the methods of image SR reconstruction are mainly divided into three types: interpolation-based methods [5], reconstruction-based methods [6], and learning-based methods [7]. The first methods such as bicubic interpolation [8]. Although the algorithm is simple, due to the lack of attention to edge information, the reconstructed image will lose details. Reconstruction-based methods require prior information to constrain the reconstruction process, and when dealing with image tasks with large magnification factors, the performance of the algorithm will become poor because of the lack of prior information.

With the rapid development of computer vision in recent years, the method based on deep learning has gradually become the mainstream. More and more deep networks with excellent performance are being created. Dong et al. [9] proposed the first model to reconstruct HR images using a convolution neural network (CNN) approach, and remarkable results were achieved compared to traditional methods, but the huge computational load limited the possibility of it being able to perform well. Deconvolution [10] and sub-pixel convolution [11] methods were proposed that can realize post-sampling at the network end, which greatly reduces the computational cost. Kim et al. designed a very deep network [12], and the introduction of residual learning [13] can effectively alleviate the gradient problem and promote the information flow of the network to accelerate the convergence of the network. Zhang et al. [14] considered the correlation between feature map channels, and proposed the channel attention mechanism combined with residual learning, which allowed the network to be more inclined to learn high-frequency information. Haris et al. [15] introduced an error feed-back mechanism to obtain better reconstruction results by calculating the upper and lower projection errors. Dai et al. [16] proposed a second-order channel attention module and non-local augmented residual group structure, which realized more powerful feature representation and feature-related learning.

As a branch of image SR, remote sensing image SR, has also developed rapidly. Lei et al. [17] proposed a multi-level representation for learning remote sensing images, concatenating the results obtained after different layers of convolution and then combining these groups with a convolution layer, which can represent local details and global environment priors. RDBPN [18] was improved from DBPN [15] by replacing its downsampling unit with a simpler downscaling unit, which greatly simplified the network. Haut et al. [19] adopted residual, skip connection and parallel convolution layers with a kernel size of 1×1 to extract more informative feature and reduce the network’s information loss. Zhang et al. [20] proposed a parallel multi-scale convolution method to extract multi-scale features and combined it with a channel attention mechanism to further utilize multi-scale feature. Zhang et al. [21] replaced element-wise addition with weighted channel connections in skip connections and performed feature optimization by modeling complex high-order statistics to further refine the extracted features. Dong et al. [22] proposed a second-order, multi-scale super-resolution network to subtly capture multi-scale feature information by aggregating features from different deep learning algorithms in a single path. Xu et al. [23] combined the details of remote sensing images with the background information by connecting local and global memory to increase the receptive field. In order to speed up the calculation speed of the model, the space size of the feature map is also reduced by the way of down-sampling. Aiming at the problem that traditional supervision methods struggle to obtain paired HR and LR images, Zhang et al. [24] designed a cyclic convolution neural network composed of two cyclic modules, which could be trained with unpaired data and had good robustness to image noise and blur. Li et al. [25] designed a recursive block, which focused on high-frequency information through the attention mechanism, and combined low-resolution and high-resolution hierarchical local information to reconstruct the image. Dong et al. [26] proposed a dense-sampling network, which enabled the network to jointly consider multiple levels when performing reconstruction priors and achieved good experimental results.

All the above methods achieved state-of-the-art performance at that time. However, their biggest problem is having too many parameters, which will place a heavy computational burden on hardware facilities, and make them not conducive to effective use in real life. In recent years, some scholars have also begun to focus on lightweight image SR networks that can be used in daily life. For example, take the network based on feature distillation [27,28,29]. Although the channel-separation operation can gradually expand the receptive field and extract more comprehensive information, this operation will lead to insufficient information flow between the separated channels [30], thereby hindering the expression of feature information. There is also the multi-scale feature extraction network MADNet, having the attention mechanism proposed by Lan et al. [31], which using repeated feature extraction blocks not only makes the network’s structure redundant, but also increases the number of network parameters.

In the field of remote sensing, it is very unreasonable to obtain high-quality images by spending more on high-precision sensors, especially in some specific fields of application, such as field surveying, individual reconnaissance, and vehicular satellite positioning and navigation. Most of the devices they use are portable, placing higher requirements on the weight of remote sensing image SR algorithms. Therefore, in order to build a network with fewer parameters and more competitive performance, we propose a lightweight feature distillation and enhancement network (FDENet). The parameters number only 501K, which is almost half the number in the advanced MADNet, and the experimental performance was also better. Figure 1 gives the overall architecture of FDENet. We exploit the backward fusion module (BFM) [32] to fuse the features extracted by four cascaded FDEBs, and then use the Gaussian context transformer (GCT) [33] to improve the feature-expression ability. FDEB contains two parts: the feature-distillation part and the feature-enhancement part. The feature-distillation part uses channel separation to extract layered features. To avoid the problem of insufficient information expression caused by this operation, we use the proposed stepwise fusion mechanism (SFM) to fuse features retained after stepwise distillation to promote information flow. The bilateral bands in the feature-enhancement part are used to enhance feature and extract complex background information from remote sensing images. Finally, the features of the two are fused to enhance the feature expression.

Overall, the main contributions are as follows:We propose a shallow pixel attention block (SRAB), which introduces the pixel attention mechanism, which can make the network pay attention to repair the missing texture details with very few parameters.We propose the SFM, which fuses the retained feature after stepwise distillation to make full use of the reserved features and promote information flow, so that it can make the feature expression more comprehensive.We propose a bilateral feature-enhancement module (BFEM), which extracts contextual information and enhances the resulting feature separately by means of a bilateral band.

## 2. Proposed Method

In this part, we will introduce the structure of our proposed network and then introduce the proposed feature-distillation part and feature-enhancement part in detail.

### 2.1. Network Architecture

FDENet’s overall structure is shown in Figure 1. We first extract primary features from the LR image, then extract deep features through four cascaded FDEBs, and finally, pass the data through a 3×3 convolution layer and an upsampling layer to obtain the SR image.

(1) Primary feature extraction: Given a LR image $I_{\mathrm{LR}}\in R^{H\times W\times 3}$, where *H*, *W*, and 3 are the length, width, and number of channels, to make the network as lightweight as possible, we only use a 3×3 convolution layer for primary feature extraction, and let $F_{\mathrm{init}}\left(\ldots \right)$ denote a convolution layer with a kernel size of 3×3 and the number of channels *C*. Then, the obtained primary feature $F_{0}$ is:(1)$$F_{0}=F_{\mathrm{init}}\left(\mathrm{LR}\right)\in R^{H\times W\times C}$$

(2) Deep feature extraction: We use four lightweight cascaded FDEBs to extract deep feature. Let $F_{{\mathrm{FDEB}}_{i}}\left(\ldots \right)$ and $F_{\mathrm{GCT}}$ denote the feature generated after passing through the *i*th FDEB block and the feature generated after passing through the GCT module, respectively, where $i\in \left[1,4\right]$. Then, the output deep feature $F_{d}$ is:(2)$$F_{d}=F_{\mathrm{GCT}}\left(F_{{\mathrm{FDEB}}_{4}}\left(\ldots F_{{\mathrm{FDEB}}_{i}}\left(\ldots F_{{\mathrm{FDEB}}_{1}}\left(F_{0}\right)\ldots \right)\ldots \right)\right)\in R^{H\times W\times C}$$

(3) Reconstruction layer: Let $F_{\mathrm{up}}\left(\ldots \right)$ denote a convolution layer with a kernel size of 3×3 and an upsampling layer. Then, the final reconstructed SR image $I_{\mathrm{SR}}$ is:(3)$$I_{\mathrm{SR}}=F_{\mathrm{up}}\left(F_{0}+F_{d}\right)$$

### 2.2. The Proposed FDEB

The proposed FDEB consists of a feature-distillation part and a feature-enhancement part. Next, we will give more details.

#### 2.2.1. Feature-Distillation Part

The structure of feature-distillation part is shown in the blue box in Figure 2 right. First, we use a convolution layer with a kernel size of 3×3 to extract the features roughly, and then use the method of stepwise distillation to increase the receptive field and further extract the layered features. Specifically, we use the stepwise channel separation operation to retain some of the features and extract information from the other portion. However, the channel-separation operation of stepwise feature distillation inevitably leads to insufficient information flow between channels, hindering the expression of features. Thus, we propose the SFM, which fuses the features retained after each distillation and uses the SRAB to extract information. This can not only make full use of the retained feature, but also effectively avoids the problem of insufficient information flow between channels. We take the proposed SRAB (as shown in Figure 3) as the basic unit of FDEB feature extraction. On top of the SRB proposed by RFDN [29], we introduce the pixel attention mechanism, which enables the network to focus on repairing the missing textural details when extracting features. Let the input feature of the nth FDEB be $F_{i}^{n}$; then, the output feature $F_{D}^{n}$ of this process can be described as:(4)$$\left\{\begin{array}{ccccc} F_{{\mathrm{refined}}_{1}}^{n},F_{{\mathrm{coarse}}_{1}}^{n}={\mathrm{split}}_{1}^{n}\left(\mathrm{Conv}3\left(F_{i}^{n}\right)\right) & \; & & \; & \\ F_{{\mathrm{refined}}_{2}}^{n},F_{{\mathrm{coarse}}_{2}}^{n}={\mathrm{split}}_{2}^{n}\left({\mathrm{SRAB}}_{{\mathrm{coarse}}_{1}}\left(F_{{\mathrm{coarse}}_{1}}^{n}\right)\right)\; & \; & & \; & \\ F_{{\mathrm{refined}}_{3}}^{n},F_{{\mathrm{coarse}}_{3}}^{n}={\mathrm{split}}_{3}^{n}\left({\mathrm{SRAB}}_{{\mathrm{coarse}}_{2}}\left(F_{{\mathrm{coarse}}_{2}}^{n}\right)\right)\; & \; & & \; & \\ F_{{\mathrm{refined}}_{4}}^{n}={\mathrm{SRAB}}_{{\mathrm{coarse}}_{3}}\left(F_{{\mathrm{coarse}}_{3}}^{n}\right)\; & \; & & \; \end{array}\right.$$
(5)$$\left\{\begin{array}{cccc} F_{{\mathrm{distil}}_{1}}^{n}={\mathrm{SRAB}}_{{\mathrm{refined}}_{1}}\left(\mathrm{concate}\left(F_{{\mathrm{refined}}_{1}}^{n},F_{{\mathrm{refined}}_{2}}^{n}\right)\right) & \; & & \\ F_{{\mathrm{distil}}_{2}}^{n}={\mathrm{SRAB}}_{{\mathrm{refined}}_{2}}\left(\mathrm{concate}\left(F_{{\mathrm{distil}}_{1}}^{n},F_{{\mathrm{refined}}_{3}}^{n}\right)\right)\; & \; & & \\ F_{D}^{n}=\mathrm{Conv}1\left(\mathrm{concate}\left(F_{{\mathrm{refined}}_{4}}^{n},F_{{\mathrm{distil}}_{2}}^{n}\right)\right)⊕F_{i}^{n}\; & \; & \end{array}\right.$$
(6)$${\mathrm{SRAB}}_{i}=\sigma \left(\mathrm{Conv}1\left(F_{i}\right)\right)⊗\mathrm{SilU}\left(\mathrm{Conv}3\left(F_{i}\right)⊕F_{i}\right)$$

Equations (Equation 4)–(Equation 6) constitute the feature distillation process used to extract the stratified features, the SFM, and the general formula of SRAB, respectively. $F_{i}^{n}$ represents the input features of the nth FDEB; $F_{{\mathrm{coarse}}_{i}}^{n}$ and $F_{{\mathrm{refined}}_{i}}^{n}$ represent the ith distillation feature and the ith retained feature in the nth FDEB, respectively. ${\mathrm{split}}_{j}^{n}$ represents the jth operation of channel separation in the nth FDEB, SRAB represents our shallow pixel attention block, $F_{i}$ represents the input features of the corresponding ${\mathrm{SRAB}}_{i}$, $F_{{\mathrm{distil}}_{i}}^{n}$ represents the retained features after fusing and extracting features, and $F_{D}^{n}$ represent the output features of this whole process.

#### 2.2.2. Bilateral Feature Enhancement Module

Compared with natural images, remote sensing images have more complex structural and background information. Therefore, in order to make full use of its background information, we propose the BFEM (as shown in Figure 4), which can focus on extracting the background information of remote sensing image while enhancing features. Let $F_{n}$ represent the input features for the enhancement block. Then, the output features $F_{i}^{n+1}$ of this process are:(7)$$F_{i}^{n+1}=\mathrm{Conv}1\left(concate\left(F_{{\mathrm{BFEM}}_{\mathrm{up}}},F_{{\mathrm{BFEM}}_{\mathrm{down}}}\right)\right)$$
where $\mathrm{Conv}1\left(\ldots \right)$ represent the 1*1 convolution layer; $\mathrm{concate}\left(\ldots \right)$ represents the operation of fusing features; $F_{{\mathrm{BFEM}}_{\mathrm{up}}}$ and $F_{{\mathrm{BFEM}}_{\mathrm{down}}}$ represent the output features of upper and lower sidebands, respectively.

In the upper sideband, we use the enhanced spatial attention [34] (ESA) to expand the receptive field extracted by FDEBs and help to obtain a clearer reconstructed image. This part is composed of a step convolution layer with a step size of 2 and a kernel size of 3×3; a maxpooling layer with a step size of 3 and a kernel size of 7×7; and three convolution layers with a kernel size of 3×3. Let $\hat{F}$ represent the features obtained through these above steps and $F_{D}^{n}$ represent the input features for the upper sideband. Then, the feature $F_{{\mathrm{BFEM}}_{\mathrm{up}}}$ obtained after passing the upper sideband can be expressed as:(8)$$F_{{\mathrm{BFEM}}_{\mathrm{up}}}=F_{D}^{n}⊗\sigma \left(\mathrm{Conv}1\left(\mathrm{Conv}1\left(F_{D}^{n}\right)⊕\hat{F}\right)\right)$$
where ⊗ and ⊕ represent element-wise multiplication and element-wise summation, respectively. $\mathrm{Conv}1\left(\ldots \right)$ represents the convolution layer with a kernel size of 1×1. σ represents the sigmoid function.

The lower sideband is used to extract the contextual feature information of remote sensing images to help obtain more details from the complex background. This part is composed of an avgpooling layer with step size of 2, a kernel size of 2×2, a convolution layer with a kernel size of 1×1 and a bilinear upsampling layer. Let $\tilde{F}$ represent the features obtained through these above steps and $F_{D}^{n}$ represent the input features of the lower sideband. Then, the feature $F_{{\mathrm{BFEM}}_{\mathrm{down}}}$ obtained after passing the lower sideband can be expressed as:(9)$$F_{{\mathrm{BFEM}}_{\mathrm{down}}}=F_{D}^{n}⊗\sigma \left(\mathrm{Conv}1\left(\mathrm{Conv}1\left(F_{D}^{n}\right)⊕\tilde{F}\right)\right)$$

### 2.3. Gaussian Context Transformer

The structure of GCT as shown in Figure 5, compared with some other attention mechanism, it is not only lighter, but also can achieve context feature motivation, leading to better performance. Therefore, we pass the features through the GCT to improve the features’ expression ability before sampling.

## 3. Results

### 3.1. Experimental Settings

#### 3.1.1. Dataset

DIV2K [35] is a dataset of 900 natural images with a resolution of 2K. It includes various natural images of buildings, animals, plants, etc. Following SMSR [26], we chose the first 800 images as the training set and the last 100 images as the validation set. Following the example of FeNet [30], we randomly selected 240 images from the UC Merced dataset containing 21 scenes to make two test sets, RS-1 and RS-2 [36]. Both of them contain 120 images and cover for the composite evaluation. RS-1 contains 120 images from ten classes, including agricultural, airplane, baseballdiamond, beach, buildings, chaparral, denseresidential, forest, freeway, and golfcourse (12 images per class). The RS-2 contains 120 images from ten classes, including intersection, mediumresidential, mobilehomepark, overpass, parkinglot, river, runway, sparseresidential, storagetanks, and tenniscourt (12 images per class). In order to further prove the generalization ability of the proposed model, we also tested it on four natural benchmark datasets—Set5 [37], Set14 [38], Urban100 [39], and BSD100 [40].

#### 3.1.2. Degradation Method

We used the bicubic interpolation method to downsample the original high-resolution image ×2, ×3, and ×4 in MATLAB R2018a to obtain LR images as training and test data.

#### 3.1.3. Training Details

We chose the L1 loss function [41] as the training loss function, which calculates the sum of the absolute difference between the actual value and the target value. Let ${\hat{y}}^{i}$ represent the SR image and $y^{i}$ represent the real HR image. Then, the loss function can be expressed as:(10)$$L_{1}\left(y^{i},{\hat{y}}^{i}\right)=\frac{1}{m}\sum_{i=0}^{m}\left|y^{i}-{\hat{y}}^{i}\right|$$

In order to get the most out of training data, we used random rotation and flipping to enhance the data. The randomly cropped training patch size of the HR image was 192 × 192, and we set the pixel range of the input image to between [0, 1]. The ADAM [42] was used as an optimizer with $\beta_{1}=0.9$, $\beta_{2}=0.999$; the initial learning rate was set to $5\times 10^{-4}$; and the learning rate decayed by half every 200 epochs, for a total of 500 epochs. All experiments were implemented using the pytorch framework, and we used a NVIDIA Tesla V100 GPU to complete the entire training and testing process.

#### 3.1.4. Evaluation Index

We used the peak signal-to-noise ratio (PSNR) and structural similarity (SSIM) to evaluate the results [43]. Let *x* and *y* be the ground truth value and the reconstructed SR image, respectively. Then, the PSNR value is: (11)$$\mathrm{PSNR}\left(x,y\right)=10\log_{10}\frac{255^{2}}{\mathrm{MSE}\left(x,y\right)}$$
(12)$$\mathrm{MSE}\left(x,y\right)=\frac{1}{H\times W}\sum_{i=1}^{H}\sum_{j=1}^{W}{\left[X(i,j)-Y(i,j)\right]}^{2}$$
where *H* and *W* represent the height and width of the given image; 255 represents the maximum RGB value for each pixel; X(i,j) and Y(i,j) represent the sizes of the pixels corresponding to the the real HR image and generated SR image, respectively. The SSIM value is:(13)$$\mathrm{SSIM}\left(x,y\right)=\frac{\left(2\mu_{x}\mu_{y}\right)\left(2\sigma_{xy}+C_{2}\right)}{\left(\mu_{x}^{2}+\mu_{y}^{2}+C_{1}\right)\left(\sigma_{x}^{2}+\sigma_{y}^{2}+C_{2}\right)}$$
(14)$$\sigma_{xy}=\frac{1}{N-1}\sum_{i=1}^{N}\left(x_{i}-\mu_{x}\right)\left(y_{i}-\mu_{y}\right)$$
where $\mu_{x}\left(\mu_{y}\right)$ and $\sigma_{x},\sigma_{y}$ represent the mean and variance, respectively; $\sigma_{xy}$ is the covariance of *x* and *y*; $C_{1}$ and $C_{2}$ are constants. We evaluate the PSNR and SSIM values on the *y* channel of the transformed YCbCr space [12]. The conversion method is as follows:(15)$$\left\{\begin{array}{ccccc} Y=0.257\times R+0.564\times G+0.098\times B+16\; & \; & & \; & \\ \mathrm{Cb}=-0.148\times R-0.291\times G+0.439\times B+128\; & \; & & \; & \\ \mathrm{Cr}=0.439\times R-0.368\times G-0.071\times B+128 \end{array}\right.$$
where *Y*, Cb, and Cr represent the brightness, the difference between the blue part of the input signal and the brightness value of the RGB signal, and the difference between the red part of the input signal and the brightness value of the RGB signal, respectively.

### 3.2. Comparison of Visualization Results

#### 3.2.1. Results on Remote Sensing Images

We quantitatively compare the results of FDENet with the results published in CVPR, ECCV, TGRS, and other well-known conferences and journals over the years. It can be clearly seen in Table 1 that our model has excellent performance when the amplification coefficients are ×2, ×3, and ×4. Take the advanced FeNet [30] as an example. Our PSNR values on the data sets RS-1 and RS-2 are 0.02 and 0.09 dB higher. On the whole, we have fewer parameters and multi-adds than most models. Figure 6 and Figure 7 are the visualization results of each model on the remote sensing image datasets. Take Figure 6 as an example. Compared with the advanced FeNet, the white car and red car in our reconstructed image have a clearer outline and more comprehensive details. From other comparison graphs, we can also see that the edge details of our model’s result graph are also richer.

#### 3.2.2. Results on Natural Images

In order to further prove the generalization ability of our model, we compared it with remote sensing image SR models on four natural benchmark datasets, Set5 [37], Set14 [38], BSD100 [39], and Urban100 [40]. Table 2 shows our quantitative comparison’s results. It can be seen in the table that our model still performs better on natural images than other SR models of remote sensing images. Although its performance for ×2 magnification is slightly inferior to that of the advanced MADNet [31] and FeNet [30], it is far superior on all data sets of ×3 and ×4. Figure 8 shows the visualization results of each model under the ×3 magnification factor on BSD100 [39] and Urban100 [40] datasets, from which we can see that after our model’s reconstruction, the window shape is better restored and the outlines are clearer; more details are retained.

## 4. Discussion

### 4.1. Comparison of SRB and SRAB

Our feature extractor’s basic unit, SRAB, introduces the pixel attention mechanism, which has been proven suitable for lightweight networks and can repair the missing texture details of images in the feature-extraction process [32]. The pixel attention mechanism only uses a convolution layer with a kernel size of 1×1 and a sigmoid function to obtain the attention maps, and then they will be multiplied with the input features. Due to this method being used in the feature-distillation part, the number of channels of the feature map will gradually decrease during the distillation process, so the amount of parameters introduced can be almost ignored. Table 3 shows that under the same conditions, the results of our FDENet after using SRAB are better than using SRB on four test sets. This fully proves the effectiveness of the SRAB.

### 4.2. Comparison of ESA and BFEM

Since the background information of remote sensing images is more important than that of nature images, it contains a variety of complex scenes, and the scales of features in different scenes are not the same. Therefore, we propose the BFEM, which can focus on extracting the contextual information of remote sensing images. Compared with the ESA proposed by RFANet [34], the BFEM proposed by us adds a lower sideband for extracting context information, to avoid introducing a large number of parameters. Instead of using a large convolution kernel, we use an avgpooling layer, a bilinear upsampling layer, and some convolution layers with a kernel size of 1*1 to achieve this goal. Table 4 is the result of FDENet replacing the feature-enhancement module with an upsampling factor of four for ESA [34] and BFEM. Under the same conditions, we can see that our BFEM has only 37K more parameters than ESA, showing more powerful performance on four test sets.

### 4.3. Analysis of SFM

The SFM we proposed fuses the reserved features after each extraction and extracts the features through SRAB, which not only makes full use of the reserved features, but also alleviates the problem of insufficient information flow in the process of feature extraction. Table 5 shows the results of our ablation experiment on SFM. Due to our SFM adopting the strategy of fusing reserved features, we are always in a very light state when using reserved features. This is not only due to the effectiveness of SFM itself, but also due to the ability of our SRAB to effectively extract feature. From the table, we can see that our SFM only introduces 9K parameters, and the PSNR values for the four data sets were 0.03, 0.01, 0.01, and 0.09 dB higher, respectively.

### 4.4. Analysis of Model Complexity

The quantity of parameters is an important indicator for evaluating the quality of lightweight models. From the results shown in Table 1 and Table 2, although our number of parameters exceeds those of SRCNN [9], LGCNet [17], and the advanced lightweight model FeNet [30], our performance is far ahead of theirs, which fully makes up for the shortcoming of more parameters. In a comprehensive comparison, our number of parameters was still shown to be less than those of most models, and our performance is more competitive. In addition to evaluating the complexity of the model with parameter quantity, we also used multi-adds to evaluate the computational complexity of the network. We set the size of the query image (HR image) to 1280 × 720. Compared with some recent models, such as IDN [27], LESRCNN [44], and MADNet [31], FDENet also has relatively few multi-adds.

## 5. Conclusions

In this article, we proposed a lightweight feature distillation and enhancement network for SR tasks of remote sensing images. Specifically, we proposed a SFM that can effectively alleviate the problem of insufficient information flow caused by channel separation during feature distillation. We use the designed lightweight SRAB as the main feature extraction method of the FDEB, which can make the network more inclined to extract high-frequency details when extracting features without introducing a large number of parameters. After feature extraction, we enhance the features with SFM, which can extract background information of remote sensing images while enhancing the features. A large number of experiments showed that our model has strong competitiveness compared with some advanced models in terms of performance and parameter quantity. This provides a certain application foundation for lightweight remote sensing image super-resolution reconstruction in field investigations, individual reconnaissance, and other fields of application.

## Figures and Tables

**Figure 1 sensors-23-03906-f001:**
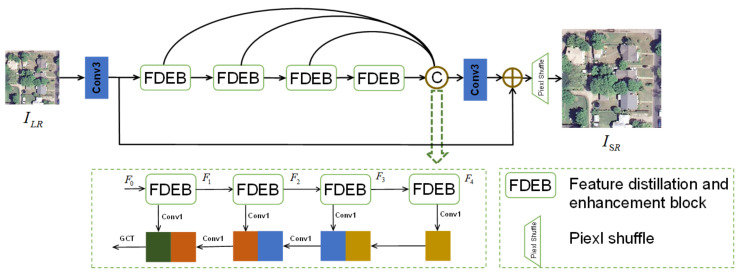
The overall network architecture of the proposed FDENet. The content in the green box represents the backward fusion module; ⊕ represents the element-wise summation.

**Figure 2 sensors-23-03906-f002:**
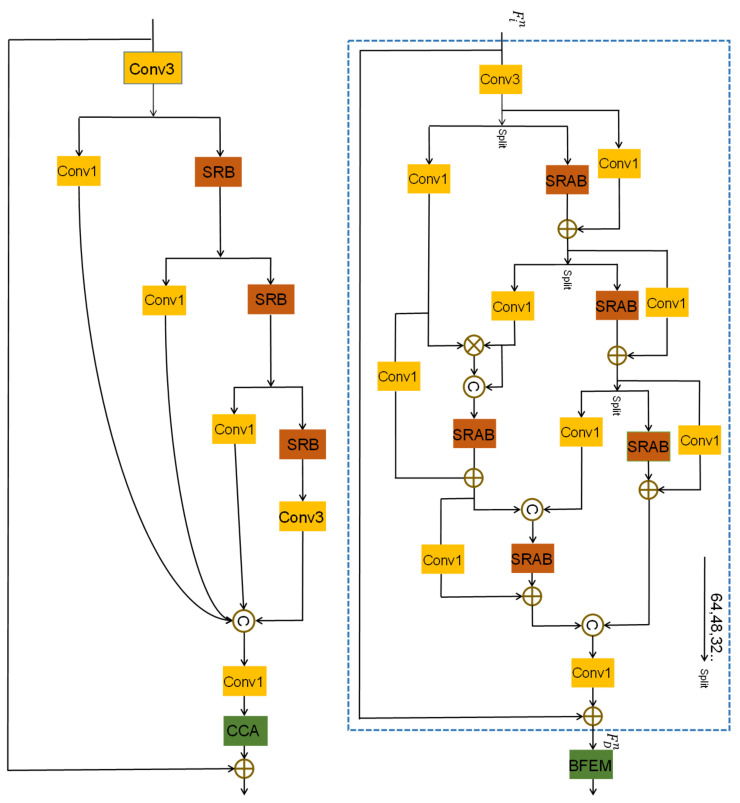
Comparison between the RFDB and the FDEB. (**Left**), the structure of RFDB. (**Right**), the structure of our FDEB. *©* represents the feature fusion; ⊕ and ⊗ represent the element-wise summation and the element-wise summation multiplication, respectively. The green and brown boxes represent the basic feature-extraction unit and the feature-enhancement block, respectively.

**Figure 3 sensors-23-03906-f003:**
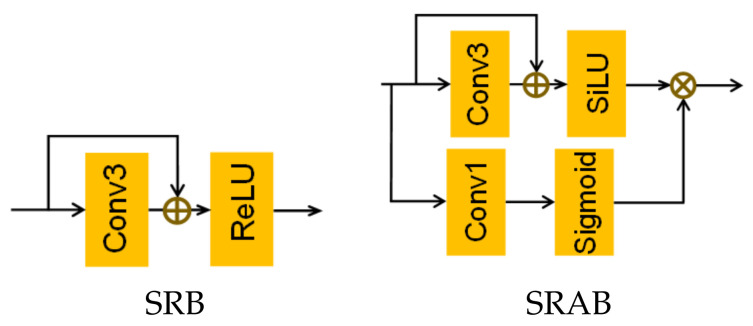
The comparison between SRB and SRAB. (**Left**), the structure of SRB. (**Right**), the structure of the SRAB; ⊕ and ⊗ represent the element-wise summation and the element-wise summation multiplication, respectively.

**Figure 4 sensors-23-03906-f004:**
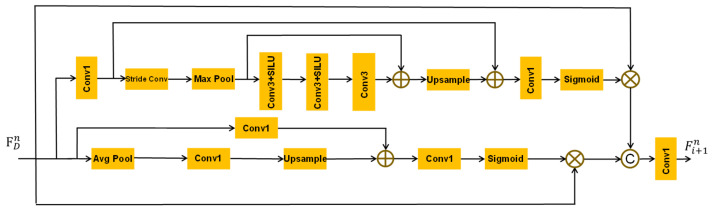
The structure of our well-designed bilateral bands. The upper sideband is used to enhance the features, and the lower sideband is used to extract the complex background information. *©* represents the feature fusion; ⊕ and ⊗ represent the element-wise summation and the element-wise summation multiplication, respectively.

**Figure 5 sensors-23-03906-f005:**
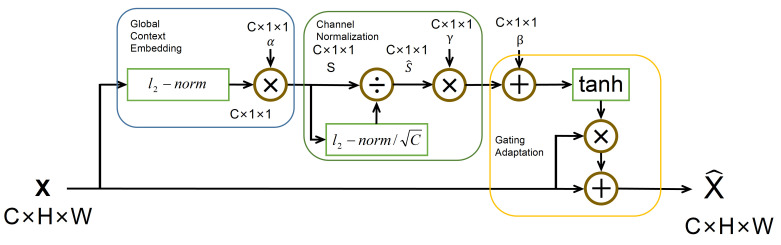
The structure of the Gaussian context transformer (GCT).

**Figure 6 sensors-23-03906-f006:**
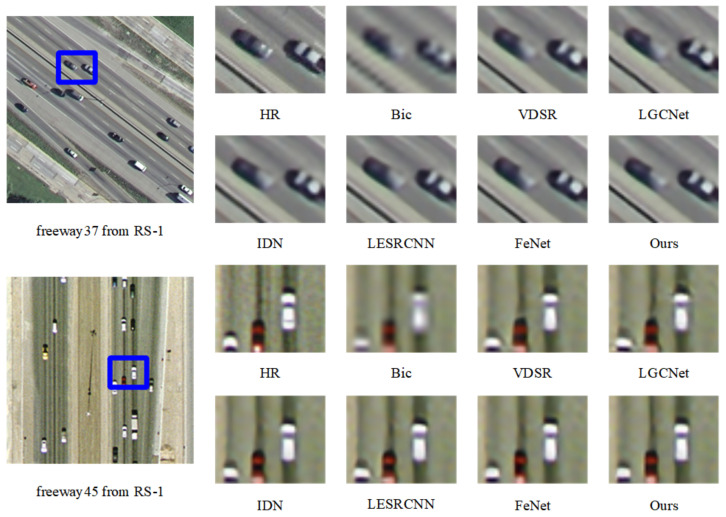
Visualization results of several SR methods and our proposed network, FDENet, on the RS-1 dataset for ×3 SR. Zoom in with blue box for best view.

**Figure 7 sensors-23-03906-f007:**
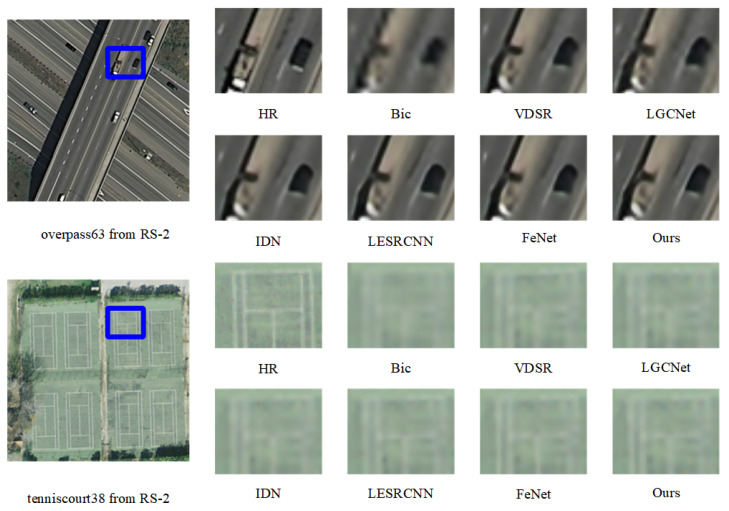
Visualization results of several SR methods and our proposed network, FDENet, on the RS-2 dataset for ×3 SR. Zoom in with blue box for best view.

**Figure 8 sensors-23-03906-f008:**
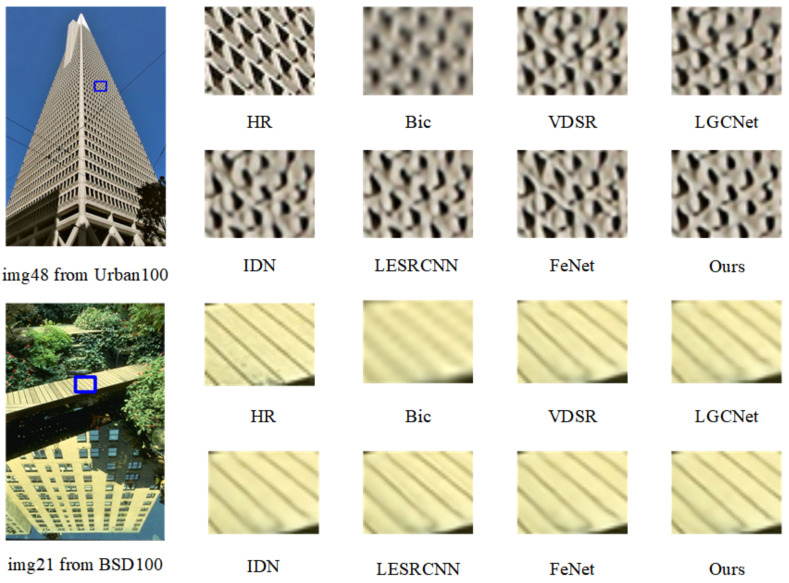
Visualization results of several SR methods and our proposed network, FDENet, on natural datasets for ×3 SR. Zoom in with blue box for best view.

**Table 1 sensors-23-03906-t001:** Quantitative evaluation results of remote sensing data sets. “Params” represents the model parameter quantity; the best and second best results are red and blue, respectively. “-” indicates that no result is provided.

Method	Scale	Params	RS-T1	RS-T2
			PSNR/SSIM	PSNR/SSIM
Bicubic		-	33.25/0.8934	30.64/0.8837
SRCNN [9]		57 K	35.18/0.9243	32.87/0.9209
VDSR [12]		666 K	35.85/0.9312	33.86/0.9312
LGCNet [17]	×2	193 K	35.65/0.9298	33.47/0.9281
IDN [27]		55 3K	36.13/0.9339	34.07/0.9329
LESRCNN [44]		626 K	36.04/0.9328	34.00/0.9320
FeNet [30]		351 K	36.23/0.9341	34.22/0.9337
FDENet (ours)		480 K	36.26/0.9346	34.28/0.9338
Bicubic		-	29.73/0.7818	27.23/0.7697
SRCNN [9]		57 K	30.95/0.8228	28.59/0.8180
VDSR [12]		666 K	31.55/0.8352	29.40/0.8391
LGCNet [17]	×3	193 K	31.30/0.8314	29.03/0.8312
IDN [27]		553 K	31.73/0.8430	29.59/0.8450
LESRCNN [44]		810 K	31.68/0.8398	29.65/0.8444
FeNet [30]		357 K	31.89/0.8432	29.80/0.8481
FDENet (ours)		488 K	31.98/0.8488	29.88/0.8489
Bicubic		-	27.91/0.6968	25.40/0.6770
SRCNN [9]		57 K	28.87/0.7382	26.46/0.7296
VDSR [12]		666 K	29.33/0.7546	27.03/0.7525
LGCNet [17]	×4	193 K	29.13/0.7481	26.76/0.7426
IDN [27]		553 K	29.56/0.7623	27.31/0.7627
LESRCNN [44]		774 K	29.62/0.7625	27.41/0.7646
FeNet [30]		366 K	29.70/0.7688	27.45/0.7672
FDENet (ours)		501 K	29.72/0.7658	27.54/0.7697

**Table 2 sensors-23-03906-t002:** Quantitative results of four super-resolution benchmark datasets. “Params” and “Multi-Adds” represent the model’s parameter quantity and model complexity, respectively. The best and second-best results are red and blue, respectively. “-” indicates that no result was provided.

Method	Scale	Params	Multi-Adds	Set5	Set14	B100	Urban100
				PSNR/SSIM	PSNR/SSIM	PSNR/SSIM	PSNR/SSIM
Bicubic		-	-	33.66/0.9299	30.24/0.8688	29.56/0.8431	26.88/0.8403
SRCNN [9]		57 K	52.7 G	36.66/0.9542	32.45/0.9067	31.36/0.8879	29.50/0.8946
VDSR [12]		666 K	612.6 G	37.53/0.9587	33.03/0.9124	31.90/0.8960	30.76/0.9140
LGCNet [17]		193 K	178.1G	37.31/0.9580	32.94/0.9120	31.74/0.8939	30.53/0.9112
SRMDNF [45]		1513 K	347.7 G	37.79/0.9600	33.32/0.9150	32.05/0.8980	31.33/0.9200
IDN [27]	×2	553 K	124.6 G	37.83/0.9600	33.30/0.9148	32.08/0.8985	31.27/0.9196
LESRCNN [44]		626 K	281.5 G	37.65/0.9586	33.32/0.9148	31.95/0.8964	31.45/0.9206
MADNet [32]		878 K	187.1 G	37.94/0.9604	33.46/0.9167	32.10/0.8988	31.74/0.9246
FeNet [30]		351 K	77.9 G	37.90/0.9602	33.45/0.9162	32.09/0.8985	31.75/0.9245
FDENet (ours)		480 K	138.7 G	37.89/0.9594	33.50/0.9170	32.15/0.8988	32.02/0.9270
Bicubic		-	-	30.39/0.8682	27.55/0.7742	27.21/0.7385	24.46/0.7349
SRCNN [9]		57 K	52.7 G	32.75/0.9090	29.30/0.8215	28.41/0.7863	26.24/0.7989
VDSR [12]		666 K	612.6 G	33.66/0.9213	29.77/0.8314	28.82/0.7976	27.14/0.8279
LGCNet [17]		193 K	79.0 G	33.32/0.9172	29.67/0.8289	28.63/0.7923	26.77/0.8180
SRMDNF [45]		1530K	156.3 G	34.12/0.9250	30.04/0.8370	28.97/0.8030	27.57/0.8400
IDN [27]	×3	553 K	56.3 G	34.11/0.9253	29.99/0.8354	28.95/0.8013	27.42/0.8359
LESRCNN [44]		810 K	238.9 G	33.93/0.9231	30.12/0.8380	28.91/0.8005	27.70/0.8415
MADNet [31]		930 K	88.4 G	34.26/0.9262	30.29/0.8410	29.04/0.8033	27.91/0.8464
FeNet [30]		357 K	35.2 G	34.21/0.9256	30.15/0.8383	28.98/0.8020	27.82/0.8447
FDENet (ours)		488 K	61.7 G	34.28/0.9253	30.33/0.8415	29.05/0.8033	28.03/0.8494
Bicubic		-	-	28.42/0.8104	26.00/0.7027	25.96/0.6675	23.14/0.6577
SRCNN [9]		57 K	52.7 G	30.48/0.8628	27.50/0.7513	26.90/0.7101	24.52/0.7221
VDSR [12]		666 K	612.6 G	31.35/0.8838	28.01/0.7674	27.29/0.7251	25.18/0.7524
LGCNet [17]		193 K	44.5 G	30.87/0.8746	27.82/0.7630	27.08/0.7186	24.82/0.7399
SRMDNF [45]		1555 K	89.3 G	31.96/0.8930	28.35/0.7770	27.49/0.7340	25.68/0.7730
IDN [27]	×4	553 K	32.3 G	31.82/0.8903	28.25/0.7730	27.41/0.7297	25.41/0.7632
LESRCNN [44]		774 K	241.6 G	31.88/0.8903	28.44/0.7772	27.45/0.7313	25.77/0.7732
MADNet [31]		1002 K	54.1 G	32.11/0.8939	28.52/0.7799	27.52/0.7340	25.89/0.7782
FeNet [30]		366 K	20.4 G	32.02/0.8919	28.38/0.7764	27.47/0.7319	25.75/0.7747
FDENet (ours)		501 K	35.9 G	32.12/0.8929	28.52/0.7795	27.53/0.7339	25.97/0.7811

**Table 3 sensors-23-03906-t003:** Results of our model on four test sets after using SRAB and SRB, respectively.

Method	Params	RS-T1	RS-T2	BSD100	Urban100
With SRB	501K	29.68/0.7675	27.52/0.7697	27.52/0.7341	25.94/0.7815
with SRAB	501K	**29.72/0.7658**	**27.54/0.7697**	**27.53/0.7339**	**25.97/0.7811**

Notes: The best results are indicated in bold font.

**Table 4 sensors-23-03906-t004:** Results of our model on four test sets after using ESA or BFEM.

Method	Params	RS-T1	RS-T2	BSD100	Urban100
With ESA	463K	29.70/0.7656	**27.55/0.7692**	27.51/0.7335	25.88/0.7779
with BFEM	501K	**29.72/0.7658**	27.54/0.7697	**27.53/0.7339**	**25.97/0.7811**

Notes: The best results are indicated in bold font.

**Table 5 sensors-23-03906-t005:** Results of our model on four test sets based on whether or not SFM is used.

Method	Params	RS-T1	RS-T2	BSD100	Urban100
w/o SFM	492K	29.69/0.7652	27.53/0.7691	27.52/0.7335	25.88/0.7796
w/ SFM	501K	**29.72/0.7658**	**27.54/0.7697**	**27.53/0.7339**	**25.97/0.7811**

Notes: The best results are indicated in bold font.

## Data Availability

Not applicable.

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
