# Peer review of "A Lightweight Feature Distillation and Enhancement Network for Super-Resolution Remote Sensing Images"

_sensors, 2023, doi:10.3390/s23083906_

Round 1

Author Response

Thank you for your comments, please see the attachment .

Reviewer 2 Report

Critical response for the manuscript entitled “Lightweight Feature Distillation and Enhancement Network for Remote Sensing Image Super-Resolution” by Feng Gao and others

The manuscript under consideration is dedicated to important problem of remote sensing image preparation and processing. Authors tried to alleviate resolution increase problem by implementing proposed lightweight feature distillation and enhancement network. Authors proved applicability of the proposed method for SR tasks and compared its advantages relatively existing methods. However, some issues could be resolved to improve paper importance for international reader. 

The applicability of obtained SR images and limitations of the results should be explained. Which existing satellite images/products and levels of processing that technology could be applied to? Does model require to be re-trained/re-tested for that case? More information about used training and testing datasets should be given. 

Besides of the conceptual charts (Figure 1) more information should be provided to prove replicability of the results. Besides of the mentioned PyTorch library at least pseudocode could given to explain implementation of the concept (real code repository is even better). I also encourage authors to share their source code, especially when they have used open-source software for research. 

Figures 1-3 also need legends describing the meaning of terms, polygons and colors (or reference at used ISO, if applicable). Please, don’t make readers guess their meanings. 

Lines 266-277; 307-309. How did authors verified that configuration they did is optimal? Need more information about model validation process.

Conclusion. I sincerely thank authors for sharing the opportunity of reading results of their research. Manuscript worth publication and all my recommendations above are optional.

Author Response

Thank you for your comments, please see the attachment.
